# E-Agent: A Cost-Efficient Agentic Framework for Knowledge-Intensive and Reasoning Tasks

## Abstract

The deployment of large language model (LLM)-powered agents for knowledge-intensive and reasoning tasks is often prohibitively expensive, since processing large volumes of evidence incurs massive token costs. Existing techniques such as prompt compression and model routing attempt to reduce token usage, but they often compromise accuracy or fail to capture the fine-grained structure of reasoning tasks. In this work, we introduce E-Agent, a cost-effective framework that leverages the pricing asymmetry of LLMs to significantly reduce monetary cost without sacrificing performance. E-Agent adopts an executor–verifier paradigm: multiple small, locally deployed models act as executors to generate candidate answers, which are then verified by a powerful cloud-based model. This design shifts token consumption from expensive outputs to relatively cheaper inputs. The framework further supports specialized workflows for both retrieval-augmented generation (RAG) and non-RAG tasks, and employs structured outputs to minimize candidate answer length. Experiments on GSM8K, ALFWorld, HotpotQA, and StrategyQA demonstrate that E-Agent reduces token usage by 10%–50% compared with strong baselines, while maintaining or even improving accuracy.

## 1 Introduction

The rapid advancement of large language models (LLMs) (Nam et al., 2024; Ge et al., 2023; An et al., 2024) has driven the evolution of intelligent agents, yielding notable progress in question answering (Gao et al., 2025), reasoning (Yao et al., 2023a), planning (Wang et al., 2025), and interactive applications (Xu et al., 2024). With the growing scale of models, from hundreds of billions to over one trillion parameters, the cost of deploying LLM-powered agents is unlikely to decrease in the foreseeable future. This imposes high financial overhead on ordinary users who rely on agents for knowledge-intensive and reasoning tasks. Such tasks include multi-turn analytical dialogues (Deshpande et al., 2025), cross-document reasoning (Yuan et al., 2024), and retrieval-augmented generation (Wu et al., 2024b), all of which require agents to ingest and reason over large volumes of domain-specific evidence. Agents must consume massive numbers of tokens to accomplish these tasks, thereby incurring prohibitively high monetary costs.

To reduce token usage in LLM-based tasks, prior work has proposed techniques such as prompt compression (Liskavets et al., 2025), context truncation (An et al., 2024), retrieval optimization (Salemi et al., 2024), and model routing (Hu et al., 2024). Prompt compression often removes salient cues that are later needed for multi-step reasoning, and it provides no guarantee of result accuracy. Contextual truncation breaks cross-turn dependencies and often forces the model to re-derive context, which paradoxically increases total token usage over long interactions. Retrieval optimization improves evidence quality but frequently increases input length by adding passages, and it does not control when or how the model should stop generating. Model routing selects a model for the whole query or turn, which is coarse relative to the fine-grained structure of reasoning; it ignores selective re-execution and fails to track token usage across subtasks. As a result, prior methods struggle to reliably reduce token consumption without undermining result accuracy.

In this work, we propose E-Agent, a cost-effective and accurate agentic framework for knowledge-intensive and reasoning tasks. Our core design is motivated by the widely observed pricing asymmetry of LLMs: current providers charge more for output tokens than for input tokens, because the prefilling stage can more effectively utilize the underlying hardware than the autoregressive decoding stage.

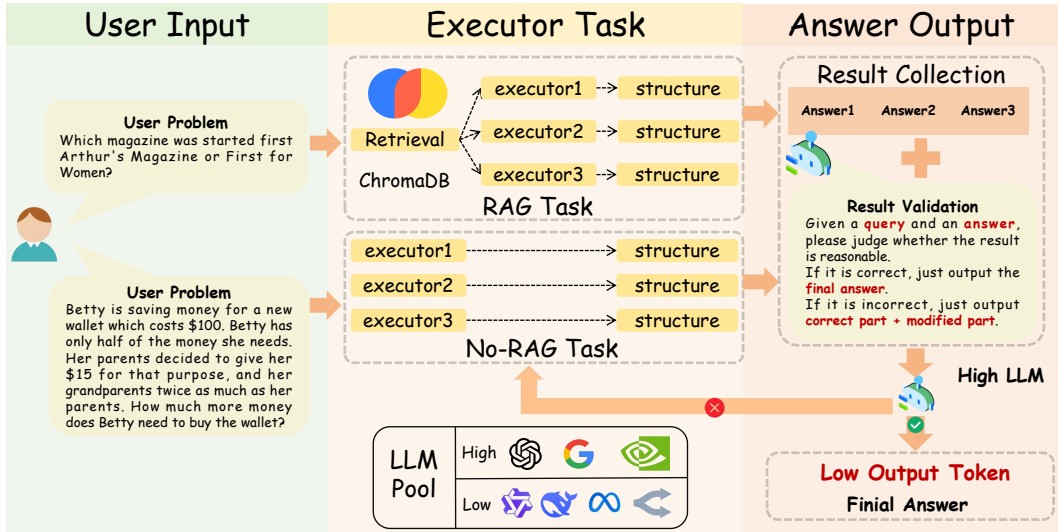

Figure 1: Overview of the Cost-Efficient Agentic framework. User queries are processed by multiple local-based small language model executors (with or without retrieval), transformed into structured outputs, and aggregated. A cloud-based large language model then validates and refines results, yielding concise final answers with reduced output tokens.

Based on this observation, we employ an executor–verifier paradigm in our framework. Multiple small, locally deployed models act as executors responsible for generating candidate answers to a given query. These candidates are then fed to a powerful but costly cloud model for verification. We design distinct workflows for retrieval-augmented generation (RAG) (Arslan et al., 2024) tasks and non-RAG tasks. With this framework, we effectively turn expensive output tokens into relatively cheaper input tokens, thereby reducing monetary cost. Although small models are less capable than a cloud model, we find that they exhibit expertise in various tasks. To compensate for the degraded accuracy of any single executor, we deploy multiple executors to simultaneously generate candidate answers. To further reduce token consumption, we enforce structured outputs instead of free-form ones from local models. This not only decreases the length of candidate answers but also improves the quality of prompts sent to the verifier, easing its reasoning process.

Experiments on GSM8K, ALFWorld, HotpotQA, and StrategyQA show that E-Agent reduces token usage by 10%–50% compared with strong baselines, while maintaining or improving accuracy and runtime. Ablations confirm that both small-model collaboration and structured outputs are indispensable, jointly enabling a superior balance of cost-efficiency and reliability.

The main contributions of this paper are as follows:

1. By leveraging the pricing asymmetry of commercial LLMs, we propose **E-Agent**, which follows an Executor–Verifier paradigm for knowledge-intensive and reasoning tasks.
2. E-Agent further reduces token consumption and helps preserve result accuracy by employing multiple executors and structured outputs.
3. We demonstrate the effectiveness of our framework on GSM8K, ALFWorld, HotpotQA, and StrategyQA, reducing token consumption by 10%–50% compared with strong baselines while maintaining or improving accuracy.

## 2 RELATED WORK

Existing studies on reducing token usage can be broadly categorized into three directions: LLM system optimization, model collaboration, and token or cost optimization.

**LLM system optimization.** Prompting methods like CoT (Wei et al., 2022), Self-Consistency (Wang et al., 2023), and Least-to-Most (Zhou et al., 2023) strengthen reasoning but expand token usage

Table 1: Quoted input and output token prices per million tokens and their output-to-input ratios. All prices are taken from official provider documentation as of September 17, 2025, and may vary by region or service tier.

| Model | Currency | Input per MTok | Output per MTok | Out/Input |
|---|---|---|---|---|
| **Doubao-Seed-1.6-thinking** | CNY | 0.8 to 2.4 | 8.0 to 24 | 10x |
| **DeepSeek-R1** | CNY | 4 | 12 | 3x |
| **Qwen3-max-preview** | CNY | 6 to 15 | 24 to 60 | 4x |
| **GPT-5** | USD | 1.25 | 10.0 | 8x |
| **Gemini 2.5 Pro** | USD | 1.25 to 2.5 | 10 to 15 | 8x to 6x |
| **Claude Sonnet 4** | USD | 3 | 15 | 5x |

through longer outputs or multiple generations. ReAct (Yao et al., 2023b) mitigates this by shifting part of the reasoning to tool calls. Retrieval-augmented generation (RAG) (Lewis et al., 2020) boosts factuality with retrieved passages via components such as DPR (Karpukhin et al., 2020) and REALM (Guu et al., 2020), though often at the cost of longer contexts.

**Model collaboration.** Multi-agent systems such as CAMEL (Li et al., 2023), AutoGen (Wu et al., 2024a), HuggingGPT (Shen et al., 2023), and ChatDev (Qian et al., 2024) improve robustness by distributing tasks across agents or tools, but usually manage tokens with heuristics. Cost-aware routing instead allocates queries between strong and cheap models, as in FrugalGPT (Chen et al., 2024) and RouteLLM (Ong et al., 2024), while sparse MoE architectures like Switch Transformer (Fedus et al., 2022) save computation by activating only a subset of experts.

**Token and cost optimization.** Another line of work explicitly controls tokens and budgets. LLMLingua (Jiang et al., 2023) and LLMLingua-2 (Pan et al., 2024) compress prompts via token selection, while MemPrompt (Madaan et al., 2022) and MemGPT (Packer et al., 2023) offload history to external memory to prevent context growth. Architectural advances such as Reformer (Kitaev et al., 2020), Longformer (Beltagy et al., 2020), BigBird (Zaheer et al., 2020), and FlashAttention-2 (Dao, 2024), along with streaming methods like StreamingLLM (Xiao et al., 2024) and RingAttention (Liu et al., 2023), further improve long-context efficiency.

In summary, prior efforts improve prompting, retrieval, memory, and collaboration, but remain fragmented and cost-unaware. Our work complements these by proposing the Cost-Efficient Agentic framework, which unifies task workflows and explicitly exploits input–output price asymmetry to reduce cost usage while preserving accuracy.

## 3 System Design

### 3.1 Pricing Asymmetry of Cloud Models

The current cloud-based large language model providers charge users differently for input tokens and output tokens. The input tokens are substantially cheaper than the output ones. As shown in Table 1, this gap is consistent across major providers: instance, Doubao-Seed-1.6-thinking shows a tenfold difference between input and output tokens, and GPT-5 maintains an 8× disparity. This imbalance stems from different computation patterns of prefilling (for input tokens) and decoding (for output tokens) stages, The prefilling stage computes attentions scores of input tokens in a highly parallel manner that utilizes GPUs effectively, while the autoregressive decoding stage must proceed sequentially, making each generated token more expensive (Zhong et al., 2024). Formally, the total monetary cost of a task can be expressed as

$$\text{Cost} = p_{in}\tau_{in} + p_{out}\tau_{out}, \quad \text{where } p_{out} > p_{in}. \tag{1}$$

$p_{in}$ and $p_{out}$ are prices charged by providers for each input and output tokens, and $\tau_{in}$ and $\tau_{out}$ are token numbers consumed in the task. Such a pricing gap suggests us a cost-effective agentic framework towards knowledge-intensive and reasoning tasks: we can employ free local models to generate candidate answers, and use powerful cloud models for validation.

Table 2: This summary of model strengths and weaknesses is derived from official provider documentation and publicly reported benchmark results.

| Model | Strengths | Weaknesses |
|---|---|---|
| **Phi-4-mini-reasoning** | simple reasoning, formatting, and information extraction | multi-hop reasoning and complex logical chains |
| **DeepSeek-R1-Distill-Qwen-7B** | Lightweight, strong reasoning ability, and resource-efficient | weaker on complex coding tasks, factual QA, and safety robustness |
| **Meta-Llama-3-8B-Instruct** | general-purpose reasoning and instruction following | fine-grained reasoning and highly specialized tasks |
| **Qwen3-32B-AWQ** | deep semantic understanding, structured outputs, and retrieval-heavy tasks | generative flexibility and cross-domain generalization |

### 3.2 AN EXECUTOR-VERIFIER AGENTIC FRAMEWORK

With the pricing asymmetry, we propose a cost-effective agentic framework for knowledge-intensive and reasoning tasks, following an **Executor–Verifier** paradigm. In this framework, we deploy $n$ small local models $m_l^i, i \in [1,n]$, which are less capable but free, serving as executors. The executors are responsible for generating candidate answers to input tasks. These candidate answers are then verified by a powerful but costly cloud model $m_c$. Ideally, the cloud model completes the task by generating only a few tokens to indicate the selection of candidate answers.

Figure 1 provides an overview of our agentic framework. We divide knowledge-intensive and reasoning tasks into two distinct workflows: one requiring retrieval-augmented generation (RAG) and one without RAG. Specifically, given a query $q$ in a RAG task, we first retrieve $k$ supporting textual segments $s_j, j \in [1,k]$ from a vector database. The query, combined with the retrieved segments, is then sent to the executors to generate candidate answers as

$$A_i = m_l^i(q, \bigcup_{j=1}^{k} s_j),\ i \in [1,n]. \tag{2}$$

The answers are then sent to the cloud model for verification as

$$(V, A^*) = m_c(q, \bigcup_{i=1}^{n} A_i, \bigcup_{j=1}^{k} s_j), \tag{3}$$

where $V$ is the verification result and $A^*$ is a corrected answer. If the proper answer appears in the candidates, $V$ indicates its number and $A^*$ has no content. If none of candidate answers are proper, the cloud model summarizes supporting segments itself and outputs the corrected answer $A^*$.

For non-RAG tasks, we directly input the query $q$ to local models for generating candidate answers

$$A_i = m_l^i(q, \bigcup_{i=1}^{n} A_i'),\ i \in [1,n], \tag{4}$$

where $\bigcup_{i=1}^{n} A_i'$ are wrong answers from the previous round, and at the first round there is no content. The answers generated by local models are pipelined to the cloud model for verification

$$V = m_c(q, \bigcup_{i=1}^{n} A_i). \tag{5}$$

$V$ also indicates which candidate answer is correct if the verification passes. However, if the verification fails, we re-feed the $q$ to local models as well as the wrong answers as shown in Equation 4. The local models are expected to generate higher-quality answers with this reflection workflow (Renze & Guven, 2024).

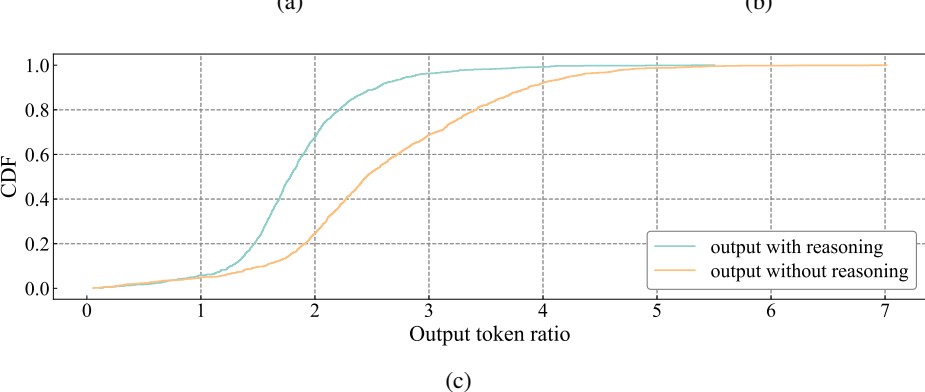

(a)                                                              (b)

(c)

Figure 2: (a)Structured compression. (b)Structured evidence. (c)CDF of token ratios between unstructured and structured outputs. Structured settings include output with reasoning and output without reasoning. Results are obtained using the `Qwen3-32B-AWQ` model on 1000 randomly sampled QA pairs from the GSM8K dataset, with statistics computed solely over output tokens.

### 3.3 LOCAL MODEL EXPERTISE

The small models deployed locally are inevitably not as capable as cloud models from providers. A key challenge of our framework, therefore, is whether local models can generate qualified answers without the massive parameters of cloud models. We observe that although small pre-trained models are not as powerful or versatile as cloud models, they often exhibit strengths in specific tasks. For example, Qwen3-32B-AWQ is reliable for generating structured outputs, while Llama-3-8B-Instruct performs well on reasoning tasks. Table 2 summarizes the strengths and weaknesses of the local models we deploy in this work.

To address this limitation, we employ a multi-executor design, in which the same query is fed to $n$ local models for generating candidate answers, as described in Equations 2 and 4. As long as one executor generates a correct answer, the cloud model can verify it, as shown in Equations 3 and 5.

### 3.4 STRUCTURED OUTPUTS

For knowledge-intensive and reasoning tasks, generating free-form responses often leads to verbose and redundant outputs. This increases not only the monetary cost but also the difficulty of the subsequent verification stage using cloud models. To address this, we enforce local models to generate only structured outputs in JSON/Schema format. With structured outputs, we find that substantially fewer tokens are generated by local models, while answer quality is largely preserved. This greatly reduces the number of input tokens billed by the cloud model. Figure 2(a) shows an example of a free-form response *vs.* a structured response. We randomly sample 1,000 queries and compare the output length with and without structured outputs. The cumulative distribution function (CDF) of the compression ratio is shown in Figure 2(c). The results show that without reasoning, 75% of responses are compressed by at least 2× and 31% by at least 3×. Even with reasoning, 37%

of responses are still compressed by at least 2×. Another advantage of structured outputs is that they form explicit relations between statements and evidence. Figure 2(b) illustrates an answer with its evidence sources explicitly attached, which facilitates the verification process of cloud models. By enforcing structured outputs, we not only reduce monetary cost but also improve the accuracy of cloud-based verification.

# 4 EXPERIMENTS

This section presents the experimental setup, including datasets, implementation details, and baseline configuration, followed by empirical results that evaluate the proposed framework.

## 4.1 DATASETS

To assess performance across varying levels of reasoning difficulty, we evaluate on four benchmarks spanning arithmetic problem solving, interactive environments, and open-domain question answering. These datasets jointly cover explicit multi-step reasoning and implicit strategy-based inference: **GSM8K** (Cobbe et al., 2021) for grade school math problems requiring numerical reasoning, **ALF-World** (Shridhar et al., 2021) for embodied agents in text-based environments, **HotpotQA** (Yang et al., 2018) for multi-hop reasoning across Wikipedia, and **StrategyQA** (Geva et al., 2021) for yes/no questions requiring implicit strategies such as temporal or causal inference. Detailed dataset statistics are provided in Section A.1.

## 4.2 IMPLEMENTATION

To reflect the large–small collaboration design, we adopt a two tier model setup. For high capability models, we use **Doubao-Seed-1.6-thinking-250715** and **DeepSeek-R1-0528**, which act as planners and validators in complex reasoning tasks. For low capability models, we deploy **meta-llama/Meta-Llama-3-8B-Instruct**, **Qwen/Qwen3-32B-AWQ**, and **microsoft/Phi-4-mini-reasoning** locally via vLLM (Kwon et al., 2023) and Sglang (Zheng et al., 2024), assigning them to subtask execution and result compression. Since the low-capability models are deployed locally, their runtime cost is mainly electricity. To bound this factor, we estimate the upper limit assuming three NVIDIA RTX 3090 GPUs operate continuously for one hour per task. With a typical power draw of 350W per GPU and an average electricity price of 0.54 CNY/kWh, the cost is below 0.57 CNY per task. In practice, GPUs do not sustain peak power throughout execution, and task completion usually requires far less than one hour in Table 3, so the actual cost is substantially lower. This value is negligible relative to commercial API charges, and we therefore omit local inference cost in aggregate accounting without affecting the validity of comparisons.

Unless otherwise specified, all experiments adopt fixed decoding parameters with temperature set to 0.7, top-$p$ to 0.9, and a maximum generation length of 8192 tokens. Implementations are based on `LangChain`, `AutoGen` and executed on a server equipped with an Intel Xeon Platinum 8352V CPU at 2.10 GHz and 1024 GB of memory.

## 4.3 BASELINES

**Input Only prompting (IO)**: A single pass baseline that requests only the final answer. It minimizes token cost but fails on multi step reasoning and interactive tasks.

**Chain of Thought (CoT)** (Wei et al., 2022): Produces stepwise rationales before the answer, boosting compositional reasoning at the expense of longer outputs.

**ReAct** (Yao et al., 2023b): Interleaves reasoning and actions for interpretability and flexibility, but token usage grows with accumulated histories.

**MAReAct** (Liu et al., 2024): Summarizes past interactions into distilled facts, cutting context length while preserving state.

**SuRe** (Kim et al., 2024): Compresses retrieved documents into summaries, reducing input tokens while keeping essential evidence.

| Method | GSM8K | | | | ALFWorld | | | | HotpotQA | | | | StrategyQA | | | |
|---|---|---|---|---|---|---|---|---|---|---|---|---|---|---|---|---|
| | In | Out | Total | Acc | In | Out | Total | Acc | In | Out | Total | Acc | In | Out | Total | Acc |
| *(a) Doubao-Seed-1.6-thinking-250715* | | | | | | | | | | | | | | | | |
| IO | 1.02 | 0.25 | 2.81 | 78.19 | 3.06 | 0.46 | 6.13 | 61.24 | 3.18 | 1.02 | 10.71 | 63.25 | 0.55 | 0.23 | 2.28 | 71.54 |
| | ↓ 5.53, ↓ 50.51% | | | | ↓ 1.99, ↓ 51.76% | | | | ↓ 5.34, ↓ 36.42% | | | | ↓ 2.35, ↓ 31.24% | | | |
| CoT | 2.38 | 0.36 | 4.78 | 81.32 | 6.26 | 0.53 | 9.25 | 62.34 | 4.58 | 1.24 | 13.59 | 65.32 | 0.94 | 0.28 | 2.99 | 71.62 |
| | ↓ 2.40, ↓ 15.92% | | | | ↓ 0.89, ↓ 27.17% | | | | ↓ 3.27, ↓ 19.29% | | | | ↓ 2.27, ↓ 12.60% | | | |
| ReAct | 3.91 | 0.56 | 7.60 | 83.56 | 9.94 | 0.89 | 15.07 | 63.02 | 5.99 | 1.85 | 19.59 | 66.14 | 1.19 | 0.47 | 4.71 | 73.87 |
| | ↓ 0.16, ↑ 28.08% | | | | ↓ 0.21, ↑ 18.61% | | | | ↓ 5.53, ↑ 16.36% | | | | ↓ 0.02, ↑ 32.19% | | | |
| MAReAct | 5.78 | 0.51 | 8.71 | 83.69 | 11.23 | 0.84 | 15.70 | 63.56 | 8.28 | 1.87 | 21.59 | 67.32 | 1.43 | 0.54 | 5.46 | 73.46 |
| | ↓ 0.03, ↑ 52.97% | | | | ↑ 0.33, ↑ 23.62% | | | | ↓ 2.45, ↑ 28.21% | | | | ↓ 0.43, ↑ 51.67% | | | |
| SuRe | 6.21 | 0.53 | 9.20 | 81.53 | 13.10 | 0.93 | 17.92 | 62.36 | 13.69 | 1.93 | 26.39 | 66.25 | 1.52 | 0.57 | 5.78 | 73.67 |
| | ↓ 2.19, ↑ 61.76% | | | | ↓ 0.87, ↑ 41.08% | | | | ↓ 2.34, ↑ 56.72% | | | | ↓ 0.22, ↑ 59.79% | | | |
| **E-Agent** | **4.02** | **0.31** | **5.69** | **83.72** | **10.68** | **0.52** | **12.70** | **63.23** | **11.25** | **0.98** | **16.84** | **68.59** | **1.25** | **0.31** | **3.48** | **73.89** |
| *(b) DeepSeek-R1-0528* | | | | | | | | | | | | | | | | |
| IO | 1.03 | 0.26 | 7.24 | 78.23 | 3.01 | 0.47 | 17.68 | 60.14 | 3.28 | 0.98 | 24.88 | 62.58 | 0.63 | 0.24 | 5.40 | 70.13 |
| | ↓ 4.66, ↓ 65.12% | | | | ↓ 2.88, ↓ 59.70% | | | | ↓ 2.78, ↓ 39.07% | | | | ↓ 3.99, ↓ 42.30% | | | |
| CoT | 2.34 | 0.35 | 13.56 | 80.12 | 6.45 | 0.52 | 32.04 | 62.89 | 4.26 | 1.21 | 31.56 | 63.85 | 0.92 | 0.32 | 7.52 | 70.69 |
| | ↓ 2.77, ↓ 34.68% | | | | ↓ 0.13, ↓ 26.98% | | | | ↓ 1.51, ↓ 22.72% | | | | ↓ 3.43, ↓ 19.65% | | | |
| ReAct | 4.25 | 0.53 | 23.36 | 81.78 | 9.87 | 0.94 | 50.76 | 62.78 | 6.02 | 1.75 | 45.08 | 64.25 | 1.26 | 0.53 | 11.40 | 72.86 |
| | ↓ 1.11, ↑ 12.52% | | | | ↓ 0.24, ↑ 15.67% | | | | ↓ 1.11, ↑ 10.38% | | | | ↓ 1.26, ↑ 21.79% | | | |
| MAReAct | 6.12 | 0.54 | 30.96 | 82.34 | 11.25 | 0.86 | 55.32 | 62.98 | 8.01 | 1.63 | 51.60 | 64.36 | 1.92 | 0.59 | 14.76 | 73.69 |
| | ↓ 0.55, ↑ 49.13% | | | | ↓ 0.04, ↑ 26.07% | | | | ↓ 1.00, ↑ 26.34% | | | | ↓ 0.43, ↑ 57.69% | | | |
| SuRe | 5.99 | 0.53 | 30.32 | 82.67 | 12.98 | 0.91 | 62.84 | 61.69 | 11.36 | 1.82 | 67.28 | 65.13 | 2.04 | 0.64 | 17.20 | 73.87 |
| | ↓ 0.22, ↑ 46.05% | | | | ↓ 1.33, ↑ 43.21% | | | | ↓ 0.23, ↑ 64.74% | | | | ↓ 0.25, ↑ 69.23% | | | |
| **E-Agent** | **4.35** | **0.28** | **20.76** | **82.89** | **9.71** | **0.42** | **43.88** | **63.02** | **7.27** | **0.98** | **40.84** | **65.36** | **1.08** | **0.42** | **9.36** | **74.12** |

Table 3: Input (In) and Output (Out) denote token usage (in millions) and Total (Total) denotes the overall cost (in RMB). Accuracy (Acc) is also reported. The first arrow indicates the change in accuracy relative to our method, while the second arrow indicates the change in total cost relative to our method. Values highlighted in ↑ indicate improvements, and values in ↓ indicate degradations.

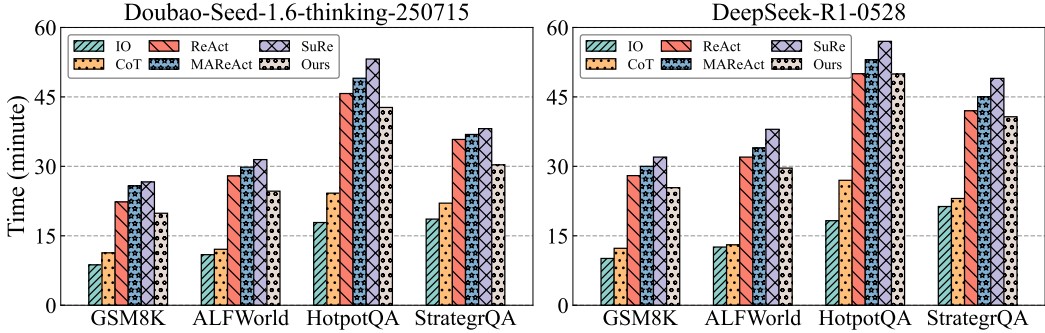

Figure 3: End-to-end runtime of different methods on GSM8K, ALFWorld, HotpotQA, and StrategyQA using Doubao-Seed-1.6-thinking and DeepSeek-R1. The reported time is measured from task loading to final answer generation, including all orchestration and model invocation steps.

## 4.4 Accuracy–Token Trade-off

Table 3 and Figure 3 present accuracy, token usage (In, Out, Total), and runtime across four reasoning benchmarks with Doubao-Seed-1.6-thinking and DeepSeek-R1.

Compared with interaction-heavy baselines (ReAct, MAReAct, SuRe), E-Agent cuts total token usage by 15–50 percent while matching or surpassing accuracy. On StrategyQA with DeepSeek-R1, SuRe

| Method | GSM8K | ALFWorld | HotpotQA | StrategyQA |
|---|---|---|---|---|
| *(a) Doubao-Seed-1.6-thinking-250715* | | | | |
| Meta | 82.23 | 62.01 | 67.26 | 71.71 |
| Qwen3 | 83.52 | 63.04 | 68.39 | 73.57 |
| microsoft | 83.45 | 62.99 | 68.48 | 73.41 |
| Meta+Qwen3 | 83.55 | 62.93 | 68.33 | 73.67 |
| Meta+microsoft | 83.49 | 63.12 | 68.32 | 73.68 |
| Qwen3+microsoft | 83.48 | 63.04 | 68.40 | 73.62 |
| Meta+Qwen3+microsoft | 83.72 | 63.23 | 68.59 | 73.89 |
| *(b) DeepSeek-R1-0528* | | | | |
| Meta | 81.79 | 60.24 | 64.16 | 72.89 |
| Qwen3 | 82.55 | 62.34 | 65.25 | 73.98 |
| microsoft | 82.56 | 63.02 | 65.30 | 73.92 |
| Meta+Qwen3 | 82.68 | 63.56 | 65.03 | 73.87 |
| Meta+microsoft | 82.81 | 62.36 | 65.19 | 73.97 |
| Qwen3+microsoft | 82.60 | 62.36 | 65.24 | 73.89 |
| Meta+Qwen3+microsoft | 82.89 | 63.02 | 65.36 | 74.12 |

Table 4: Ablation study on small-model choices under Doubao-Seed-1.6-thinking-250715 and DeepSeek-R1-0528 across four benchmarks. Meta, Qwen3, and microsoft denote meta-llama/Meta-Llama-3-8B-Instruct, Qwen/Qwen3-32B-AWQ, and microsoft/Phi-4-mini-reasoning, respectively. Due to space limitations, full model names are not shown in the table.

consumes 17.20M tokens versus 9.36M for E-Agent, yet accuracy is lower (73.87% vs. 74.12%). On HotpotQA, E-Agent reduces cost by 10–20 percent relative to ReAct and MAReAct while delivering the best accuracy (68.59% with Doubao, 65.36% with DeepSeek). Runtime is also improved, with 5–9 minute savings over ReAct and SuRe.

Against minimalist baselines (IO, CoT), E-Agent uses more tokens due to orchestration but consistently improves accuracy. For example, GSM8K accuracy rises from 78.19% (IO) to 83.72% with Doubao, and HotpotQA improves by 1.5 points over CoT with DeepSeek. IO and CoT are faster but less reliable, whereas E-Agent balances runtime with stronger accuracy.

Overall, E-Agent is more efficient than interaction-heavy pipelines and more accurate than minimalist baselines, with the greatest gains on multi-step reasoning and evidence integration tasks.

### 4.5 Impact of Small Model Selection on Performance

Table 4 analyzes the influence of small model choices within the framework. In single model settings, Meta-Llama-3-8B-Instruct (Meta), Qwen3-32B-AWQ (Qwen3), and Phi-4-mini-reasoning (Microsoft) yield broadly comparable results when paired with a high capability cloud-base model, indicating that each model reliably handles localized execution tasks. Qwen3 achieves the strongest accuracy, especially on HotpotQA and StrategyQA, which we attribute to its stronger base reasoning and comprehension that reduce intermediate errors in multi hop and implicit inference.

Beyond individual models, pairing two small models generally improves performance over single model usage, reflecting the benefit of diversity. Different models contribute complementary strengths, including factual precision, arithmetic robustness, and linguistic fluency, which together support more reliable execution of decomposed subtasks. The Meta+Qwen3 and Meta+Microsoft configurations are particularly effective and surpass their constituents in most cases.

The best results arise when all three small models are used jointly. Under both Doubao-Seed-1.6-thinking-250715 and DeepSeek-R1-0528 planners, the Meta+Qwen3+Microsoft configuration attains the highest accuracy across all benchmarks, with the largest gains on HotpotQA and StrategyQA where complex reasoning and evidence integration are required. These findings show that heterogeneous small model ensembles provide robustness and better generalization across diverse reasoning tasks, while no single small model is strictly necessary for strong end to end performance.

| Method | GSM8K | | | ALFWorld | | | HotpotQA | | | StrategyQA | | |
|--------|-------|-----|------|----------|-----|------|----------|-----|------|------------|-----|------|
| | Total | Acc | Time | Total | Acc | Time | Total | Acc | Time | Total | Acc | Time |
| *(a) Doubao-Seed-1.6-thinking-250715* | | | | | | | | | | | | |
| all cloud large model | 8.73 | 84.01 | 24.76 | 15.12 | 63.41 | 29.31 | 20.62 | 68.71 | 48.76 | 5.81 | 73.97 | 35.73 |
| w/o Structured Ouput | 6.32 | 83.73 | 23.41 | 12.93 | 63.70 | 26.73 | 17.11 | 68.55 | 44.61 | 3.62 | 73.92 | 32.91 |
| **E-Agent** | **5.69** | **83.72** | **19.87** | **12.70** | **63.23** | **24.64** | **16.84** | **68.59** | **42.72** | **3.48** | **73.89** | **30.33** |
| *(b) DeepSeek-R1-0528* | | | | | | | | | | | | |
| all cloud large model | 27.18 | 82.96 | 30.07 | 50.91 | 63.43 | 35.71 | 48.64 | 66.05 | 54.73 | 14.56 | 74.72 | 45.91 |
| w/o Structured Ouput | 25.42 | 82.83 | 28.03 | 45.81 | 63.11 | 32.93 | 44.86 | 65.47 | 48.74 | 10.67 | 74.27 | 42.74 |
| **E-Agent** | **20.76** | **82.89** | **25.39** | **43.88** | **63.02** | **29.62** | **40.84** | **65.36** | **49.95** | **9.36** | **74.12** | **40.71** |

Table 5: Total denotes the overall cost (in RMB), Acc is the accuracy, and Time is the runtime in minutes.

## 4.6 ABLATION STUDY: PERFORMANCE OF DIFFERENT COMPONENTS

In this ablation study, we compare two configurations that remove individual components against E-Agent. Overall, E-Agent consistently achieves the lowest total cost across all three benchmarks, while maintaining comparable or even higher accuracy and significantly reducing runtime in Table 5.

Among the components, relying solely on the cloud large model leads to the most substantial cost increase. For example, on DeepSeek-R1-0528, removing collaboration raises the total cost on GSM8K from 20.76 to 27.18, and on HotpotQA from 40.84 to 48.64. This highlights that small models can effectively substitute large models in certain stages, thereby reducing token consumption.

The removal of Structured Output results in moderate increases in both cost and runtime. For instance, on Doubao-Seed-1.6-thinking-250715, GSM8K runtime rises from 19.87 minutes to 23.41 minutes, and HotpotQA from 42.72 minutes to 44.61 minutes. This demonstrates that structured outputs play a key role in suppressing token usage at the output layer, which in turn improves overall efficiency and runtime performance.

These results collectively indicate that both components, collaborative execution with small models and structured outputs, are indispensable: the former curbs reliance on costly cloud decoding, while the latter minimizes output length and accelerates inference. Their combined effect enables E-Agent to achieve a superior position on the quality–cost tradeoff curve, demonstrating the necessity of integrating both mechanisms for practical efficiency.

## 5 CONCLUSION

In this work, we introduced the Cost-Efficient Agentic framework for knowledge-intensive and reasoning tasks under asymmetric pricing. By shifting computation to the inexpensive input side and constraining expensive outputs to short, verifiable conclusions, E-Agent directly addresses the economic inefficiencies of current LLM usage. The framework unifies retrieval-augmented and non-retrieval workflows, systematically integrates multiple small models through executor parallelism and verifier aggregation, and enforces structured outputs to compress redundancy while preserving verifiability. Extensive experiments across diverse benchmarks confirm that E-Agent achieves substantial token savings without sacrificing accuracy, consistently outperforming competitive baselines on the quality–cost tradeoff. These findings highlight that explicitly modeling pricing asymmetry is not only practical but also essential for building sustainable, scalable, and reliable research agents.

**Limitations and Future Work.** Our study is limited to text-only settings with static pricing, leaving multimodal toolchains, dynamic pricing, and preference adaptation for future work. Promising directions include extending orchestration to balance cost, latency, and risk; generalizing structured memory to multimodal or executable forms such as code, SQL, or graphs; expanding validation with uncertainty-aware sampling and selective rollbacks; and evaluating long-term cost–quality–latency trade-offs under real-world traffic. We aim to establish cost as a first-class design principle for next-generation LLM agents.

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

APPENDIX

# A    EXPERIMENTAL DETAILS

## A.1    DATASET

We conduct experiments on four widely used benchmarks that evaluate numerical reasoning, embodied planning, multi-hop question answering, and commonsense reasoning.

**GSM8K** (Cobbe et al., 2021) contains approximately 8.5k grade school math word problems. Each instance requires multi step numerical reasoning and is commonly solved through chain of thought style decomposition, making it a standard benchmark for mathematical reasoning with LLMs.

**ALFWorld** (Shridhar et al., 2021) is a text based environment for embodied agents with thousands of interactive tasks across six task types. Agents receive high level goals and interact through textual observations and actions, testing planning and sequential reasoning in grounded simulations.

| Dataset | Example |
|---------|---------|
| **GSM8K** | **Question**: Natalia sold clips to 48 of her friends in April, and then she sold half as many clips in May. How many clips did Natalia sell altogether in April and May? **Answer**: 72. |
| **ALFWorld** | **Question**: You are in the middle of a room. Looking quickly around you, you see a drawer 2, a shelf 5, a drawer 1, a shelf 4, a sidetable 1, a drawer 5, a shelf 6, a shelf 1, a shelf 9, a cabinet 2, a sofa 1, a cabinet 1, a shelf 3, a cabinet 3, a drawer 3, a shelf 11, a shelf 2, a shelf 10, a dresser 1, a shelf 12, a garbagecan 1, a armchair 1, a cabinet 4, a shelf 7, a shelf 8, a safe 1, and a drawer 4. Your task is to: put some vase in safe. **Actions**: go to shelf 6; take vase 2 from shelf 6; go to safe 1; open safe 1; put vase 2 in/on safe 1. |
| **HotpotQA** | **Question**: Which magazine was started first Arthur's Magazine or First for Women? **Answer**: Arthur's Magazine |
| **StrategyQA** | **Question**: Would someone in Mumbai refer to Solanum melongena as an eggplant? **Answer**: False. |

Table 6: Example instances from GSM8K, ALFWorld, HotpotQA, and StrategyQA.

**HotpotQA** (Yang et al., 2018) comprises roughly 113k questions that emphasize multi hop reasoning over Wikipedia. Each question is paired with supporting sentences, requiring retrieval and integration of evidence across documents under distractor or full wiki settings.

**StrategyQA** (Geva et al., 2021) provides yes or no questions designed to test implicit multi step reasoning. Solving these questions typically requires combining unstated facts and strategies such as temporal or causal inference, challenging models to leverage background knowledge with hidden reasoning chains.

To illustrate the data characteristics, Table 6 shows an example instance from each dataset.

## A.2 HYPERPARAMETERS

All experiments are conducted under consistent decoding settings unless otherwise specified. For both LLM-L and LLM-S we set temperature to 0.7 and top-$p$ to 0.9, with a maximum generation length capped at 8192 tokens.

For locally deployed models, prefix caching and chunked prefill are enabled to reduce latency, while remote code execution is trusted to ensure compatibility with model-specific implementations. When supported, reasoning and structured output modes are activated; otherwise, models fall back to plain text generation.

To further accelerate large-scale evaluation, we launch 128 concurrent processes for parallel inference across GPUs. These optimizations allow us to maintain stable throughput while preserving reproducibility.

## A.3 ENVIRONMENT SETUP

All experiments are conducted on a local server equipped with 3×NVIDIA RTX 3090 GPUs (24GB memory each). The system runs driver version 575.51.03 with CUDA 12.9. Inference is managed with vLLM v0.9.0.1 and Sglang (`sglang[all]` = 0.5.3rc0) for efficient scheduling.

## A.4 COST ACCOUNTING

Token usage is tracked through raw input and output counts before compression or aggregation. Prices are converted to RMB based on the official API billing rate of each provider at the time of experimentation. For models with asymmetric pricing, costs are reported separately for input tokens, output tokens, and total expenditure.

## B Ethics Statement

This work complies with the ICLR ethical guidelines. No human subjects or animal experiments were involved in this research. All datasets used were obtained in accordance with the relevant usage policies, ensuring no violation of privacy. No personally identifiable information was used, and no experiments were conducted that could raise privacy or security concerns.

## C Reproducibility Statement

All datasets used in this paper are publicly available: GSM8K, HotpotQA, StrategyQA, and ALF-World. For retrieval-augmented tasks, we construct a local ChromaDB vector store, where all documents are pre-segmented into passages of 200 tokens with an overlap of 50 tokens. For the ALFWorld environment, we adopt the official GitHub toolkit, and each task is executed with a maximum of 50 steps. A task is considered successful if it is completed within this limit.

We use meta-llama/Meta-Llama-3-8B-Instruct, Qwen/Qwen3-32B-AWQ, and microsoft/Phi-4-mini-reasoning as local executors, and Doubao-Seed-1.6-thinking-250715 and DeepSeek-R1-0528 as cloud verifiers. Hyperparameters are fixed across experiments: temperature $= 0.7$, top-$p = 0.9$, and maximum generation length $= 8192$ tokens. In multi-executor settings, we deploy $n = 3$ executors in parallel unless otherwise noted.

All reported results are averaged over three runs with random seeds 0, 1, 2. Experiments are conducted on a server equipped with $3\times$ NVIDIA RTX 3090 GPUs, an Intel Xeon Platinum 8352V CPU @ 2.10 GHz, and 1024 GB RAM. Cloud API calls are fully logged with token usage reports to enable accurate cost analysis.

## D The Use of Large Language Models

Large language models (LLMs) were used to assist in drafting and polishing the manuscript. Specifically, we employed an LLM to improve language, enhance readability, and ensure clarity across different sections of the paper. The model supported tasks such as rewriting sentences, checking grammar, and improving the overall fluency of the text.

It is important to note that the LLM was not involved in ideation, research methodology, or experimental design. All research concepts, methods, and analyses were developed and conducted solely by the authors. The contribution of the LLM was limited to improving the quality of the language and did not extend to scientific content or data analysis.

