# OpenReview forum: "E-Agent: A Cost-Efficient Agentic Framework for Knowledge-Intensive and Reasoning Tasks"
_ICLR.cc/2026/Conference — ICLR 2026 Conference Withdrawn Submission_

### Official Review · Reviewer_kpJQ · 2025-10-29

**Soundness:** 3
**Presentation:** 3
**Contribution:** 2
**Rating:** 4
**Confidence:** 4

**Summary:**

This paper introduces E-Agent, a framework designed to reduce the monetary cost of deploying LLM-powered agents for knowledge-intensive and reasoning tasks. The key insight is to exploit the pricing asymmetry between input and output tokens in commercial LLMs (where output tokens cost 3-10× more than input tokens). E-Agent uses an executor-verifier paradigm: multiple small, locally-deployed models generate candidate answers (cheap/free), which are then verified by a powerful cloud model that ideally only outputs a selection indicator (expensive but minimal). The framework enforces structured outputs to further compress token usage and supports both RAG and non-RAG workflows. Experiments on GSM8K, ALFWorld, HotpotQA, and StrategyQA demonstrate 10-50% cost reduction while maintaining or improving accuracy.

**Strengths:**

1. The pricing asymmetry observation is insightful and the proposed solution directly addresses a real deployment concern.
2. Four diverse benchmarks (arithmetic, embodied agents, multi-hop QA, strategy QA) with two different cloud models provide reasonable coverage.
3. Table 4 effectively demonstrates that heterogeneous executor ensembles outperform individual models, and Table 5 shows both components (collaboration + structured outputs) are necessary.
4. Unlike pure compression methods, E-Agent maintains or improves accuracy while reducing costs.
5. Detailed experimental setup, hyperparameters, and dataset descriptions in the appendix.

**Weaknesses:**

1. The paper mainly combines existing ideas — model ensembles, structured generation, and verification — but doesn’t really offer new theoretical insight. There’s no clear analysis of *why* or *when* this approach should work, or a principled way to decide which executors to use.

2. Local GPU costs are brushed off a bit too quickly. For teams without existing GPU infrastructure, that’s a real expense. There’s also no discussion of latency — since E-Agent needs *n + 1* model calls instead of just one. And the cloud API cost comparison is based on a single snapshot from September 2025, even though prices can fluctuate quite a lot.

3. The method only works under certain conditions — when small models can generate decent candidates, when verification is easier than generation, when you have access to local GPUs, and when tasks have structured outputs. That rules out creative writing, hard reasoning tasks where small models fail, and latency-sensitive applications.

4. There’s no comparison with simpler baselines like using a single cloud model but with a shorter output or lower temperature. It also barely compares to newer routing methods, and doesn’t include recent prompt compression approaches like RECOMP or Selective Context.

5. There’s no statistical significance testing, even though decoding is stochastic. The “all-cloud large model” baseline in Table 5 isn’t clearly defined — are the prompts the same? And the paper doesn’t really dig into what kinds of errors E-Agent introduces or avoids. The reflection mechanism for non-RAG tasks also feels under-evaluated.

6. All the benchmarks are short and in English. There’s nothing on long-context tasks (say, 100k+ tokens) where cost really becomes critical. And since the approach relies on structured outputs, it might not generalize to open-ended or creative tasks. Also, results might not hold up as model pricing or capabilities evolve.

7. Table 3 is a bit overwhelming — it’s hard to see the relationship between cost savings and accuracy. In some cases, cost goes down *and* accuracy goes up, but there’s no explanation why. It would help to discuss where E-Agent actually *doesn’t* make sense to use.

**Questions:**

1. What happens when all executors produce incorrect answers? How does verifier performance degrade? Can you provide statistics on how often this occurs?
2. E-Agent requires n+1 sequential model calls. How does end-to-end latency compare to baselines? Is there a cost-latency Pareto frontier analysis?
3. How sensitive are results to executor choice? Is there a principled way to select executors for a new task, or is this manual tuning?
4. What exactly is the prompt used for the verifier? How does it know when all candidates are wrong?
5. How does performance change with number of executors (n=1,2,3,4,5)? What about with different cloud models?
6. How would E-Agent perform on tasks requiring creative generation or very long contexts?
7. In Table 5, why does "all cloud large model" cost more than any baseline in Table 3? What's different about this configuration?
8. How much does performance degrade if structured outputs are not possible (e.g., for creative writing)?

---

### Official Review · Reviewer_u63t · 2025-10-31

**Soundness:** 2
**Presentation:** 2
**Contribution:** 2
**Rating:** 2
**Confidence:** 4

**Summary:**

The paper introduces E-Agent, a cost-efficient agentic framework for knowledge-intensive and reasoning tasks. It follows an executor–verifier paradigm, where multiple local-based executors generate candidate answers that are subsequently validated by a cloud-based verifier. Experiments on GSM8K, ALFWorld, HotpotQA, and StrategyQA demonstrate that E-Agent reduces token usage by 10%–50% compared with strong baselines such as ReAct, while maintaining accuracy or even improving it.

**Strengths:**

1. Cost-efficient agentic architecture exploiting pricing asymmetry
The proposed E-Agent framework explicitly leverages the input–output pricing asymmetry of commercial LLMs to reduce overall token expenditure by shifting computation from expensive output decoding to cheaper input prefilling, achieving 10%–50% cost savings across benchmarks.
2. Experimental evidence (Table 3) is compelling
 E-Agent consistently reduces aggregate token and monetary costs by a meaningful margin (10–50%) relative to strong baselines, while accuracy is maintained or improved. These results are corroborated by runtime measurements (Figure 3).

**Weaknesses:**

1. Lack of Methodological Originality

The proposed E-Agent framework lacks sufficient originality. Using structured outputs as inputs to cloud models to reduce token usage and improve accuracy is not a novel idea, as similar techniques have been widely adopted in agentic LLM prompt designs (e.g., A-Mem [1]). Likewise, employing multiple local LLMs for cost optimization has already been demonstrated in FrugalGPT [2]. The paper is encouraged to clarify its methodological novelty compared with these prior works.

2. Insufficient Discussion of Related Work

The “Related Work” section does not adequately compare E-Agent with closely related recent studies (e.g., KnowAgent [3]). As a result, the paper’s positioning within the landscape of cost-efficient agentic framework design remains unclear.

3. Lack of Cost-Efficient Baselines
Although several cost-reduction methods such as LLMLingua[4] and FrugalGPT[2] (which provide open-source implementations) are mentioned in the related work section, they are not included as baselines in the experiments. This omission weakens the empirical validation of E-Agent’s cost efficiency.

4. Missing Theoretical Analysis of Failure Modes

While the paper motivates cost reduction by shifting computation to input tokens, it does not model or analyze failure cases when all local executors fail. In such scenarios, the cloud verifier must regenerate answers from scratch. The frequency and impact of these failures are not quantified, nor are worst-case cost or robustness metrics provided.

5. Vague System Description

The description of the Cost-Efficient Agentic framework remains vague. The equations in Section 3.2 are mostly symbolic process representations and lack formal definitions of key implementation details such as candidate generation, structured output constraints, and task assignment. It would be helpful to elaborate on how user inputs are mapped to executor tasks and how the cloud LLM coordinates with or supervises different executors.

6. Unbalanced Experimental Baselines

Although the related work section discusses token-efficient methods like FrugalGPT[2], the experiments only compare E-Agent against reasoning-oriented baselines such as ReAct. Including additional comparisons with token-efficient agentic frameworks would strengthen the empirical evaluation and clarify the claimed advantages.

7. Minor Typographical Errors

There are small spelling mistakes in the paper, such as “Ouput” → “Output” in Table 5 and “StrategrQA” → “StrategyQA” in Figure 3.

References

[1] Wujiang Xu, Zujie Liang, and Kai Mei. A-Mem: Agentic Memory for LLM Agents. NeurIPS, 2025.

[2] Lingjiao Chen, Matei Zaharia, and James Zou. FrugalGPT: How to Use Large Language Models While Reducing Cost and Improving Performance. Trans. Mach. Learn. Res., 2024.

[3] Yuqi Zhu, Shuofei Qiao, and Yixin Ou. KnowAgent: Knowledge-Augmented Planning for LLM-Based Agents. ACL, 2025.

[4] Huiqiang Jiang, Qianhui Wu, Chin-Yew Lin, Yuqing Yang, and Lili Qiu. LLMLingua: Compressing Prompts for Accelerated Inference of Large Language Models. EMNLP, 2023.

**Questions:**

1. How does the framework ensure that executors produce structured outputs compliant with the JSON/Schema format? How does the system robustly handle cases where the output format is incorrect?
2. When the verifier needs to “regenerate an answer” rather than simply verify one, what are the common failure modes? Could you provide some examples?

---

### Official Review · Reviewer_KnxN · 2025-10-31

**Soundness:** 3
**Presentation:** 3
**Contribution:** 2
**Rating:** 4
**Confidence:** 3

**Summary:**

Given the cost imbalance between input and output tokens in LLMs, the authors propose a framework where multiple smaller models generate candidate solutions, and a larger model verifies and synthesizes the final output. This approach is evaluated on several knowledge- and reasoning-intensive datasets, achieving superior performance compared to both reasoning and non-reasoning agentic baselines.

**Strengths:**

1. The proposed idea, using small models for generation and a large model for verification, is conceptually sound and empirically supported.
2. The paper is clearly written and easy to follow.

**Weaknesses:**

1. While the paper claims to exploit input–output price asymmetry of the same model, the technical solution is based on pricing differences between small and large models. Thus, the contribution is a new collaboration framework balancing the use of small and large models, rather than directly addressing token cost asymmetry.
2. Consequently, since the contribution centers on a collaborative modeling framework, comparisons with other similar methods (e.g., speculative decoding, model routing) should be included for a fair evaluation.
3. The analysis overlooks the computational or API cost of running the smaller/local models, which should be factored into the total cost using either token prices or FLOPs.

**Questions:**

See weaknesses above

Did you use structured outputs for the baseline methods?

---

### Official Review · Reviewer_tF8W · 2025-11-01

**Soundness:** 2
**Presentation:** 2
**Contribution:** 2
**Rating:** 4
**Confidence:** 4

**Summary:**

E-Agent introduces a cost-efficient agentic framework for knowledge-intensive and reasoning tasks by exploiting the input–output pricing asymmetry of large language models (LLMs). It follows an Executor–Verifier paradigm, where multiple small local models generate structured candidate answers that are verified by a powerful cloud model, thus shifting computation from expensive output tokens to cheaper input tokens. The framework supports both retrieval-augmented and non-retrieval tasks, enforces structured outputs to reduce redundancy, and demonstrates consistent token cost reductions (10–50%) across GSM8K, ALFWorld, HotpotQA, and StrategyQA, while maintaining or improving accuracy and runtime efficiency.

**Strengths:**

1. The paper introduces a novel exploitation of LLM input–output pricing asymmetry, achieving significant monetary savings without compromising reasoning performance.
2. The multi-executor verification mechanism effectively balances small-model efficiency with large-model accuracy, improving scalability for agentic systems.
3. Enforcing structured (JSON/Schema) outputs not only reduces token consumption but also enhances verification quality and reasoning interpretability.
4. Extensive experiments across diverse benchmarks and ablations (e.g., model ensemble, structured output impact) validate both the efficiency and robustness of the approach.

**Weaknesses:**

1. Although the framework leverages pricing asymmetry, its theoretical analysis of cost modeling and optimization remains shallow, lacking a systematic mathematical or economic formulation.
2. The experiments focus mainly on text-based reasoning tasks (e.g., GSM8K, HotpotQA), without verifying the framework’s generalizability to more complex scenarios such as multimodal reasoning or code generation.
3. The framework fundamentally relies on cloud-based large models for final verification, making its cost advantage sensitive to potential API pricing or latency fluctuations.
4. The selection of executor models (e.g., Qwen, Llama, Phi) is based on empirical combinations rather than automated or theoretically grounded criteria.
5. The paper does not sufficiently analyze error propagation or performance degradation caused by inconsistencies between executors and the verifier.
6. Although the local GPU electricity cost is roughly estimated, the paper lacks a detailed evaluation of long-term, concurrent, or distributed deployment cost feasibility.

**Questions:**

See the Weaknesses.

---

### Note · Authors · 2026-01-05

I have read and agree with the venue's withdrawal policy on behalf of myself and my co-authors.